# Current Biomarker Strategies in Autoimmune Neuromuscular Diseases

**DOI:** 10.3390/cells12202456

**Published:** 2023-10-15

**Authors:** Menekse Oeztuerk, Antonia Henes, Christina B. Schroeter, Christopher Nelke, Paula Quint, Lukas Theissen, Sven G. Meuth, Tobias Ruck

**Affiliations:** Department of Neurology, Medical Faculty, Heinrich Heine University Düsseldorf, 40225 Düsseldorf, Germany; menekse.oeztuerk@med.uni-duesseldorf.de (M.O.); antoniacharlotte.henes@med.uni-duesseldorf.de (A.H.); paula.quint@med.uni-duesseldorf.de (P.Q.);

**Keywords:** CIDP, biomarkers, GBS, myasthenia gravis, neuromuscular diseases, IIM, inflammation

## Abstract

Inflammatory neuromuscular disorders encompass a diverse group of immune-mediated diseases with varying clinical manifestations and treatment responses. The identification of specific biomarkers has the potential to provide valuable insights into disease pathogenesis, aid in accurate diagnosis, predict disease course, and monitor treatment efficacy. However, the rarity and heterogeneity of these disorders pose significant challenges in the identification and implementation of reliable biomarkers. Here, we aim to provide a comprehensive review of biomarkers currently established in Guillain-Barré syndrome (GBS), chronic inflammatory demyelinating polyneuropathy (CIDP), myasthenia gravis (MG), and idiopathic inflammatory myopathy (IIM). It highlights the existing biomarkers in these disorders, including diagnostic, prognostic, predictive and monitoring biomarkers, while emphasizing the unmet need for additional specific biomarkers. The limitations and challenges associated with the current biomarkers are discussed, and the potential implications for disease management and personalized treatment strategies are explored. Collectively, biomarkers have the potential to improve the management of inflammatory neuromuscular disorders. However, novel strategies and further research are needed to establish clinically meaningful biomarkers.

## 1. Introduction: The Need for Biomarkers in Inflammatory Neuromuscular Disorders

Inflammatory neuromuscular disorders are a heterogeneous group of immune-mediated diseases with diverse underlying pathomechanisms. Epidemiology, clinical manifestations, treatment strategies, and responses vary across the spectrum of disease. Common to all is a potential severe burden of disease with conceivable long-lasting disability. Designated criteria for categorisation of affected individuals into corresponding subgroups are well-established [1]. Over the last few years, our pathophysiological understanding of autoimmune inflammatory neuromuscular disorders has steadily improved. However, essential pathogenic processes remain to be studied. In this regard, the recognition of specific biomarkers could confer additional insights while informing treatment decisions. Biomarkers are characteristic features of biological processes and are detectable and quantifiable in body fluids and tissues [2]. As valuable indicators, they serve, inter alia, diagnostic, prognostic, and therapeutic purposes in diseases.

Considering the rarity and diversity of clinical manifestation of neuromuscular disorders (NMDs), the identification of specific biomarkers for each of them is essential, particularly regarding disease course prediction and improvement of daily clinical practice. In recent years, a considerable development on this matter has emerged. However, there is still a lack of objective biomarkers suitable in NMDs.

This review has the intention to provide an overview of biomarkers currently established in Guillain-Barré syndrome (GBS), chronic inflammatory demyelinating polyneuropathy (CIDP), myasthenia gravis (MG), and idiopathic inflammatory myopathy (IIM). Our purpose is to discuss and highlight yet unmet demands concerning further parameters.

## 2. On the Concept of Biomarkers

The appliance of biomarkers has become increasingly relevant over the last decade. As useful tools, they serve various aspects in disease management. Biomarkers are indicators of both physiological mechanisms and pathogenic processes or responses to various interventions and treatment regimens in general [2,3]. Particularly in diagnostic, prognostic, and predictive aspects, biomarkers can contribute as helpful tools. The detection of the disease of interest is achieved by diagnostic biomarkers. The presence or alteration of a predictive biomarker forecasts probabilities of incidents following the exposure to an intervention or environmental factor [2]. Prognostic biomarkers aid in the estimation of clinical course and severity in the observed condition. Correspondingly, monitoring biomarkers can be employed in longitudinal disease assessment, detecting the status of a condition or measuring treatment effects. Detection of biomarkers may offer insights into causative pathomechanisms. Hence, biomarkers are crucial to the development of treatment strategies including targeted therapies, assisting healthcare for affected individuals and the population.

The further identification and utilization of biomarkers in clinical and scientific settings is essential in disease management. In this review, we intend to set up an overview of the so far published biomarkers in the mentioned autoimmune NMDs. We discuss relevant biomarkers by categorizing them into subgroups as mentioned above. Achieving a strict separation into the respective subsets is not always feasible, as certain biomarkers may serve multiple functions. Also, we aim to emphasize the prevailing unfulfilled need for the establishment of further specific biomarkers.

## 3. Biomarkers in GBS and CIDP

### 3.1. Current Biomarkers in GBS and CIDP

Few recognized biomarkers of GBS and CIDP are presently integrated in diagnostics and monitoring of disease courses and treatment responses. An overview of relevant biomarkers in use is given in Table 1.

Immune-mediated mechanisms following antecedent infections, commonly with a subset of *Campylobacter jejuni* strains with ganglioside-mimicking lipooligosaccharides (LOS), result in the typical clinical phenotype of progressive ascending symmetrical paresis of the limbs with hypo- to areflexia in GBS [174]. CIDP is an autoimmune neuropathy affecting peripheral nerves. The common clinical hallmark is the symmetrical weakness of distal and proximal portions of the limbs, whereas pure motor, pure sensory, and focal subtypes are described equally. A diagnostic delay occurs frequently in CIDP [174].

Impairments of the blood-nerve barrier and the blood-cerebrospinal fluid (CSF) barrier as barriers of the PNS are concomitant with the pathophysiology underlying GBS and CIDP. Tissue of peripheral nerves, serum, and CSF compose the predominant origins of biomarkers [25]. Biomarkers can also be linked to immediate damage of the PNS. In the following, we address the studies on barrier-, infection-, immune-, and peripheral nerve system (PNS) damage-associated biomarkers in GBS and CIDP to provide an overview of biomarkers.

### 3.2. Diagnostic Biomarkers

Elevated total protein levels in CSF with a regular white blood cell count, termed cytoalbuminological dissociation, is a common observation in GBS and CIDP [175]. The detection is one of the first steps regarding the diagnostic approach at suspicious clinical presentation. A follow-up analysis of CSF may be useful, as protein elevation in CSF is known to increase during disease duration [176,177,178].

Zhang et al. have linked GBS to low CSF index levels of fibrinogen and prealbumin with regular levels of haptoglobin [179]. Furthermore, in CIDP patients, they demonstrated normal CSF index levels of prealbumin, elevated haptoglobin, and low fibrinogen [179]. Inconsistently in other studies, haptoglobin as a plasma protein and positive acute phase protein was found increased in CSF of GBS patients [180].

GBS can be triggered by a subgroup of *Campylobacter jejuni* strains incorporating ganglioside-mimicking LOS [25,181]. Thus, infection-associated biomarkers may be informative regarding pathogenesis of GBS. LOS and serotype as well as sequence type of Campylobacter strains can be applied as diagnostic biomarkers. Islam et al. implemented genotyping of *C. jejuni* strains from affected patients [4]. Structural analyses on the *C. jejuni* LOS were performed [5]. Moreover, antibodies (Abs) against *C. jejuni* DNA-binding protein (Anti-C-Dps IgG) were detected in *C. jejuni*-related GBS cases [6]. LOS of *C. jejuni* activate innate immune responses, thus biomarkers linked to these processes may be utilized. In this regard, numerous studies have addressed the role of cytokines, complements, and chemokines (see below).

Metalloproteinases (MMPs) have recently been described as potential biomarkers for the diagnosis of CIDP, as altered levels have been found in patients with different immune-mediated disorders. MMPs are a heterogenous group of endopeptidases involved in many pathophysiological functions such as tissue destruction and infiltration by immune cells. Patients with CIDP showed elevated serum levels of MMP-9 compared to controls [7]. Increased serum levels of MMPs are not specific for CIDP and are also found in MG patients, for example. To assess the potential utility of these new biomarkers in the clinical setting, future studies are needed.

Damage of the PNS is a pathogenic component of GBS and CIDP. Certain biomarkers depicting these processes have been identified in recent years. Abs against several peripheral myelin proteins P0, P2_14–25_, PMP22, and connexin 32 have been described in GBS and CIDP patients [8,9,10,11]. In individuals who have demyelinating variants of GBS, there is a noticeable rise in IgG levels and heightened antibody response to P2 during the peak of the illness [11]. These factors could potentially play a role in the progression of the disease. However, the detection of these Abs is not widespread in clinical settings.

Increased levels of sphingomyelin (SM) in CSF of GBS and CIDP patients emerge as novel diagnostic biomarkers, as presented by Capodivento et al. [15]. SM appeared more elevated in active CIDP compared to stable disease course, highlighting its potential as a monitoring biomarker and a possible opportunity to adapt ongoing treatment strategies [15].

The cysteine protease inhibitor cystatin C is secreted from the choroid plexus into CSF. Enzyme-linked Immunosorbent Assay (ELISA) and proteomic studies revealed decrease of cystatin C in CSF of both GBS and CIDP patients [16,17,18,19]. Thus, decreased cystatin C could be applied as a biomarker for early diagnosis of GBS with limited specificity, as it has also been observed in MS patients.

Correspondingly not disease-specific, CSF 14-3-3 protein was identified in GBS patients as early as 12 to 48 h after disease commencement [21]. Detection of 14-3-3 protein as a diagnostic marker may benefit quicker disease identification and faster treatment initiation, which is especially crucial in clinical handling of GBS.

Notably, CSF interleukin-8 (IL-8) concentration seems to differentiate between CIDP and GBS, as it is high in both conditions, but significantly higher in GBS [22]. A cut-off value, as proposed by Breville et al., could aid setting both entities apart [22]. IL-8 is not disease-specific either. Nevertheless, knowledge of the mentioned relation may contribute to an early detection and enrollment of therapeutical steps.

Ultimately, it is evident that there is a lack of specificity regarding diagnostic biomarkers in GBS and CIDP. Particularly in non-bacteria-triggered disease development, reported markers are few to none. Concerning final disease determination, physicians apply data from clinical examination, electrophysiology, and CSF analysis, which may be time consuming. Detection of further diagnostic biomarkers could accelerate the start of treatment.

### 3.3. Predictive Biomarkers

Particularly in the acute stage of GBS, disease-related symptoms can develop rapidly, challenging both treating physicians and the well-being of affected individuals. In-patient stay in intensive care units is not uncommon. Thus, the knowledge of predictive biomarkers could aid further estimation of disease courses and better management of complications. Early determination of relevant biomarkers at clinical onset is crucial to initiate appropriate and strategic treatment decisions at the beginning.

Regarding damage of the myelin sheath, anti-ganglioside Abs are the most frequently communicated biomarkers. Gangliosides are sialylated glycosphingolipids located on the outer surface of nerve cells [182]. Anti-ganglioside Abs may result in complement activation and are repeatedly related to clinical phenotypes of GBS considering assorted expression of analogous antigens throughout the PNS [25,26]. IgG and IgM are the most reported subclasses of Abs to ganglioside antigens [26]. Anti-ganglioside GM1 Abs in GBS are associated with *Camplyobacter jejuni* infection and, interestingly, to clinical recovery and therapy response [27,28]. Serum anti-GQ1b IgG Abs are elevated in GBS and MillerFisher syndrome (MFS) and are closely linked to ophthalmoplegia [29]. Thus, auto-Abs to gangliosides seem to correlate with clinical phenotypes and symptom presentation, making them valuable for predictive purposes.

Similarly, efforts have been made to identify Abs against nodal and paranodal proteins. Auto-Abs against the nodal proteins neurofascin and gliomedin, leading ultimately to peripheral demyelination, are described in various studies [38,183,184]. Their presence is linked to a poorer response towards intravenous immunoglobulins (IVIg).

Earlier studies revealed Abs against the nodal and paranodal proteins neurofascin (Nfasc155 and Nfasc140/186), contactin-1 (CNTN1), contactin-associated protein-1 (Caspr1), and Caspr1/CNTN1 complex in CIDP patients [30]. These seropositive patients presented certain clinical manifestations, e.g., ataxia and tremor [31,33]. Importantly, Anti-Nfasc155 IgG4 antibody positive cases showed a younger age at disease onset and also a poorer response to IVIg treatment [34]. Anti-CNTN1 antibody-positive patients presented comparably low clinical improvement after IVIg, while benefiting from corticosteroids [35].

The detection of stated auto-Abs may be a valuable support for predictive purposes. Although the prevalence described in the literature is relatively low, integration into the diagnostic workup may help guide treatment options and improve case ascertainment. Therefore, systematical screening of auto-Abs in CIDP is advisable.

In consideration of PNS damage, biomarkers of direct neuronal disruption can be distinguished from those indicating myelin sheath damage. Neurofilaments are cytoskeletal intermediate filaments; their expression is amplified particularly in axons [39]. In GBS patients, CSF neurofilament levels, especially phosphorylated neurofilament heavy protein (pNFH) levels, were elevated and correlated positively with clinical and electrophysiological presentation [25,40,42]. Serum neurofilament light chain (Nfl) is associated with disease progression and therapy response in CIDP, accentuating its value as a predictive and prognostic biomarker [41]. Simultaneously, tau levels in CSF were increased in GBS patients and related mutually with clinical outcomes [25,42]. Tau proteins are involved in preservation of stability of microtubules throughout the nervous system [185]. Importantly, both neurofilaments and tau proteins are general indicators of structural central nervous system damage with limited specificity. In conclusion, auto-Abs to gangliosides as well as to nodal and paranodal proteins are the most relevant predictive biomarkers with comparably high occurrence.

### 3.4. Prognostic Biomarkers

Equivalent to predictive biomarkers, prognostic indicators are urgently needed to comprehend disease course. Early categorization and prognostic evaluation are essential for patients and treating physicians.

Abs against galactocerebroside (anti-Gal-C), a further component of myelin, are reportedly related to sensory deficits and autonomic disruption in GBS patients [47]. Remarkably, Gal-C-GBS is linked to antecedent Mycoplasma pneumoniae infection [47].

Anti-phospholipid Abs were reported in GBS and decreased following therapy with intravenous immunoglobulins (IVIgs) [186,187]. Neuron-specific enolase (NSE) was increased in CSF of GBS and CIDP patients and correlated with months to clinical recovery [49,50]. Importantly, neither of them exhibits disease-specificity.

Particularly in prognostic inquiries, the extensive category of cytokines plays a substantial role. Interferon gamma (IFN-γ) can potently fuel pro-inflammatory courses and correlates positively with clinical severity in GBS patients [53]. Recovery from GBS is connected to development of neutralizing auto-Abs to IFN-γ [52]. This may indicate the idea that IFN-γ can serve as a treatment target and predictive biomarker at the same time [54]. Similar observations were reported with regards to tumor necrosis factor alpha (TNF-α) [53]. IVIg treatment is associated with decreased levels of plasma TNF-α with elevated concentration of TNF-α antagonists, such as sTNFR1 [55,56]. Circulating transforming growth factor β1 (TGF β1) as an anti-inflammatory cytokine was detected decreased in the early course of GBS and downregulation appears to be linked to clinical disability [65]. CSF and plasma levels of interleukin 37 (IL-37) in GBS patients were elevated and serum levels declined after IVIg treatment [53]. Corresponding results are reported for IL-1β, IL-6, IL-12, IL-16, IL-17, IL-18, IL-22, and IL-23 [57,58,59,60,61,62]. CSF IL-17A levels were positively correlated with clinical manifestation of GBS, offering an opportunity for monitoring purposes [53]. The upregulation of IL-4 and IL-10 is linked to the subsequent recovery phase from GBS, which is associated with an improvement of clinical symptoms [14,63]. Likewise, elevation of some cytokines is reported in CIDP [64].

Importantly, serum derived from GBS patients displayed the capability to induce demyelination both in vivo and in vitro in the presence of complement components [25,188,189].

Therapeutic targeting of complement activation inhibited formation of membrane attack complex (C5b-9) and resulted in clinical improvement [54,188,189,190]. C5b-9 complexes were detected in GBS and multiple sclerosis (MS) patients [191]. Min et al. discussed probable employment of serum complement proteins as suitable biomarkers in GBS [66]. Here, C3 is linked to a higher clinical severity and longer hospitalization and is positively correlated with C3a and C5a [66]. The significance of cytokines, however, is naturally limited due to lack of specificity as well as potential alterations in the presence of comorbidities and medication.

Additional prognostic significance is attributed to the calcium-binding astroglial protein S100B. S100B serves as a glial marker and is shown to be elevated in CSF and serum of GBS patients [42,49]. Interestingly, elevated levels may be associated with clinical severity and a poor prognosis in GBS [50].

Stem cell factor (SCF) and hepatocyte growth factor (HGF) were more elevated in affected CIDP patients than in GBS patients and are considered as indicators for chronicity serving as supplementary diagnostic and monitoring biomarkers [71].

The significant drawback of prognostic biomarkers, particularly regarding cytokines, lies in their susceptibility to interference and the limited availability of specific indicators. Nevertheless, their assessment can still prove valuable in improving understanding of the disease.

### 3.5. Monitoring Biomarkers

To evaluate the individual disease progression, it is important to conduct both clinical monitoring and repetitive. Chemokines, as low-molecular-weight cytokines, and their corresponding receptors are crucial elements of inflammatory mechanisms. Increased CCR2 and monocyte chemoattractant protein 1 (MCP-1) levels were observed in GBS and levels declined in the recovery phase, revealing potential for ancillary monitoring markers [72]. CSF concentrations of CCL7, CCL27, CCR1, CCR5, CXCL10, CXCR3, CXCL9, and CXCL12 were elevated in GBS [71,73]. CXCL9, CXCL10, and CCL3 were increased in CIDP [74].

Identification of additional chemokines could provide possible targets for therapeutic agents in the treatment of GBS and CIDP. Other inflammatory mediators, among them ICAM-1, VCAM1, and VEGF were higher in CSF of GBS and CIDP patients compared to healthy controls [71].

Certainly, these described groups are not specific to the disease. However, their determination can be useful for assessing the course and at least reflect the trend.

In recent years, there has been an increasing emphasis on discovering more precise markers. Notably, significant breakthroughs have been made in the field of microRNAs, providing valuable insights.

MicroRNAs (miRNAs) are a group of short, non-coding, and single-stranded RNA molecules contributing to regulation of gene-expression. As circulating miRNAs, they are additionally present in various body fluids and are therefore detectable, giving possibility to serve as potential biomarkers and aid in understanding personalized patient analysis [77]. A study applying microarray technology and PCR analysis identified upregulated has-miR-4717-5p and has-miR-642b-5p in serum of GBS patients [78]. Regarding CIDP, only a few studies investigated miRNAs. Serum miR-31-5p was found and high levels are linked to IVIg treatment duration [80]. Regrettably, an insufficient quantity of monitoring biomarkers is presently available. Detection of miRNAs is helpful and promising but is still limited to centers equipped with suitable infrastructure. Thus, further research is necessary.

### 3.6. Required Biomarkers in GBS and CIDP

The diagnosis of GBS and CIDP are eminently challenging due to shortcomings in the identification and clinical integration of specific biomarkers. A delay in the initiation of effective immunomodulatory treatments can be the consequence. The detection of IL-8 at the beginning of clinical manifestation appears to be beneficial for disease classification. Figure 1 presents an overview of relevant biomarkers in GBS and CIDP, highlighting their significance in the respective pathogenesis. Regarding prognostic and monitoring biomarkers, chemokines and interleukins may be useful, but their increase is observed in other inflammatory processes as well, highlighting the need for more specificity. Methodological advances and unbiased analyses may help to overcome this lack. Proteomic studies showed elevated inflammation-related blood-derived proteins such as apolipoprotein A-IV, β2-microglobulin (β2-MG), vitamin D-binding protein (DBP), and α-1-antitrypsin levels in CSF of GBS patients [17,180,192,193]. Apolipoprotein E was decreased [180,194]. Transthyretin in CSF has been studied and showed disparate results with downregulation in protein profiling and upregulation when analyzed via ELISA [192,195]. Increased CSF transthyretin levels were simultaneously noticed in CIDP and MFS [195]. Their value as non-disease specific biomarkers is currently discussed in studies.

The cysteine protease inhibitor cystatin C is secreted from the choroid plexus into CSF. ELISA and proteomic studies revealed decrease of cystatin C in CSF of both GBS and CIDP patients [16,17]. It becomes evident that these described alterations also do not exhibit the necessary specificity for the disease. However, these observations, nonetheless, contribute to a better understanding of the disease onset and enable inferences to be drawn regarding mechanistic causes. Hence, further investigations are urgently required to discover disease-specific biomarkers, particularly in diagnostics.

## 4. Biomarkers in MG

### 4.1. Current Biomarkers in MG

MG is a chronic antibody-mediated autoimmune disease leading to focal or generalized muscle fatigability including respiratory symptoms or dysarthria [196,197]. Exclusive ocular symptoms (ocular MG) are possible and often represent the first clinical manifestation. Disease exacerbations inducing myasthenic crisis and ICU admission are still frequently observed. Causative auto-Abs target different components of the neuromuscular junction (NMJ) and disrupt regular transmission [198]. The prevalence is stated to be around 150–300 per million population [199]. The age of 50 years is used to distinguish between early-onset MG (EOMG) and late-onset MG (LOMG), as two peaks of incidence have been recognized [200,201]. In most cases (85%), auto-Abs against the extracellular domain of muscle nicotinic acetylcholine receptors (AChRs) are detected [84,85,86]. Biomarkers applied in MG are primarily disease-underlying auto-Abs and antigenic structures. An overview is given in Table 1. Specifically in the case of MG, the assignment of biomarkers to the initially introduced subgroups is rather ambiguous, as many functions are simultaneously fulfilled.

### 4.2. Diagnostic Biomarkers

As of now, serological testing for the detection of anti-AChR Abs is integrated in clinical practice. Anti-AChR Abs are mainly members of IgG1 and IgG3 subgroups [87]. Binding leads to heterogeneous pathogenic pathways, including the blockade of transmission and receptor internalization, followed by the formation of the membrane attack complex (MAC) [88,89,90,91,202]. Although Masuda et al. showed positive correlation between titer levels of Abs against the main immunogenic region of AChR and clinical severity [92,93], many studies, such as Vincent and Newsom-Davis, cannot confirm this coherency [197]. Anti-AChR seems to be very heterogeneous in its characteristics between individuals [203]. Differences in the specificity of AChR Abs, their ability to activate complement, the immunoglobulin subclass, or variations in serum antibody concentrations could be potential reasons for the insufficient correlation [91,105]. Using absolute serum levels of AChR Abs is not recommended to accurately predict the disease course or response to therapy in patients. The most common detection method for anti-AChR Abs is the radioimmunoprecipitation assay (RIPA), providing high specificity and sensitivity, which are described to be approximately 85% in generalized MG and 50% in ocular MG. Newer RIPAs were able to expose auto-Abs in formerly seronegative MG patients and enhanced diagnostic certainty [95,96]. Fixed cell-based assays (CBA) showed even higher sensitivity in some studies, indicating possible detection alternatives [97,98].

In 2001, Hoch et al. identified a novel determinable antibody in seronegative MG patients [100]. Muscle-specific kinase (MuSK) is a receptor tyrosine kinase required for signaling between motor neurons and skeletal muscles [102,204]. Anti-MuSK Abs mainly targets the extracellular domain of MuSK and belong mainly to the IgG4 antibody subclass [103,107]. They are detected in around 8% of MG cases and in around 30 to 50% of anti-AChR-Ab-negative MG [95,100,105]. According to AChR Abs, MuSK Abs are generally detected via RIPA [108,205]. Constant efforts are made towards developing testing alternatives, such as CBAs and ELISAs [98,206]. Utilization of CBAs has been shown to enlarge detected cases of double-positive MG patients with Abs to more than one target antigen [95,207].

Furthermore, the low-density lipoprotein receptor-related protein 4 (LRP4) contributes to NMJ upkeep as a transmembrane protein and has been described to be another target antigen in MG [111,117]. Anti-LRP4 Abs are associated primarily with IgG1 and IgG2 subclasses and initiate pathogenic complement activation [116]. Anti-LRP4-Ab-positive patients showed more pronounced clinical manifestation than in seronegative cases [114]. Prevalence of anti-LRP4 Abs is higher in women, varies widely throughout the literature, and seems to be influenced by detection assays and the investigated population [113]. Essentially, anti-LRP4 Abs are not specific for MG, as they have also been identified in serum and CSF of amyotrophic lateral sclerosis patients [112,115]. Regardless, their consideration may strengthen diagnostic definiteness.

Several other antigenic spots of interest in MG have been analyzed in the recent past, covering titin, Kv1.4 potassium channels, rapsyn, cortactin, and agrin. Their corresponding worth for diagnostic, prognostic, or monitoring ambitions has not been sufficiently clarified.

Titin Abs are detected in 20 to 40% of anti-AChR-Ab-positive patients and are tightly linked to thymoma-associated MG [95,118,120]. One limitation is their additional occurrence in MG patients over the age of 50 without presence of thymoma [208,209]. Therefore, positive testing for titin Abs in MG patients should be followed by comprehensive assessment for thymoma, particularly in patients under 50 years of age.

Kv1.4 is a voltage-gated potassium channel contributing to presynaptic acetylcholine release. Suzuki et al. identified anti-Kv1.4 Abs in sera from MG patients, associated with distinct clinical symptoms, conveying possible diagnostic and prognostic value [121,122]. Rapsyn Abs have not been identified exclusively in MG, but in several other autoimmune diseases [123]. Simultaneously, Abs against the cytoplasmic protein cortactin were encountered particularly in seronegative MG cases, but also in other immune-mediated disorders and even in healthy controls [124].

Abs against agrin, which is connected to activation of MuSK [102], have been found in formerly seronegative MG patients that turned seropositive [127]. Agrin-MG is related to EOMG, limited therapeutic response, and mild to severe clinical manifestation (see below). Thus, their quick identification could assist in clinical practice [115].

Recently, the role of free serum light chains (FLC) has been suggested as a diagnostic biomarker for MG [210]. Significantly elevated levels of FLC have been measured in MG patients via a turbidimetric assay and could be used to support the diagnosis of MG. A further association between serum FLC levels and clinical manifestations, disease severity, age at MG onset, thymoma, or treatment could not be established [210].

### 4.3. Predictive Biomarkers

In contrast to GBS and CIDP, there are no biomarkers in MG that fulfill purely predictive functions. Rather, the indicators identified for diagnostic purposes also incidentally serve predictive roles. Anti-MuSK Abs are more often detected in female patients and appear to be associated with crises and poorer outcomes [104,211,212]. In contrast to patients with anti-AChR-Abs, they additionally predict a better treatment response to rituximab [109,110]. The presence of agrin Abs correlates with a limited therapeutic response [127,128]. In clinical practice, the use of these two biomarkers with a predictive intention is relatively limited.

### 4.4. Prognostic Biomarkers

As with predictive, there are only very limited prognostic biomarkers. In this context, biomarkers tend to fulfill multiple functions simultaneously. Compatibly, anti-MuSK Ab levels are reported to correlate with disease severity, implying possible prognostic worth [106]. Since they are members of the IgG4 antibody subclass and thus do not activate the complement system, their presence could be considered as a biomarker of response to anti-CD20 therapies such as rituximab and not to complement inhibitors. Titin Abs are related to thymoma-associated MG and more frequent hospitalization [118,120]. Anti-Kv1.4. Abs associate with myasthenic crises, myocarditis, thymoma, and prolongation of QT-time [122]. While the presence of Anti-cortactin Abs is linked to rather mild clinical manifestation, Anti-agrin Abs correlate with possible severe clinical symptoms [95,126,127].

### 4.5. Monitoring Biomarkers

Monitoring in the context of Myasthenia Gravis is a delicate matter. The determination of biomarkers for this purpose is highly limited. Currently, in clinical practice, alongside regular physical examinations, clinical parameters such as vital signs and lung vital capacity play a significant orienting role. Interestingly, serum anti-AChR Abs were detected ahead of clinical manifestation of MG and even progressively elevated throughout the time, indicating possible potential as a monitoring biomarker, next to diagnostic purposes [94]. Since AChR Abs belong mainly to IgG1 and IgG3 subgroups, FcRn-targeting therapies such as efgartigimod can assist reducing pathogenic IgG autoantibody levels [213]. From this perspective, serum IgG and AChR Ab titers could also serve as useful biomarkers for monitoring treatment response in patients receiving FcRn-antagonists.

However, it is important to acknowledge that within a limited subset of patients with MG (approximately 17%), the standard RIPA using human AChR may fail to detect AChR. For these individuals, a titration method involving a combination of normal and denervated AChR might offer an opportunity to enhance assay sensitivity and diminish background precipitation. This refinement could prove valuable for affirming positive results in both monitoring and diagnostic capacities. Furthermore, one must also remain cognizant of instances where anti-AChR antibodies appear in monozygotic twins or first-degree relatives even in the absence of clinical myasthenic symptoms [198].

### 4.6. Required Biomarkers in MG

Present diagnostic tests for MG, including clinical examination, antibody analysis, and neurophysiological assessments, do not automatically mirror disease course. Consequently, solid prognostic markers are urgently required. Figure 2 provides a comprehensive view of relevant biomarkers in MG.

Observations of ongoing efforts can be noted in the identification of additional helpful indicators for MG. Serum metabolic profiles in EOMG and LOMG patients were examined by Lu et al. and revealed significant metabolite alterations between both disease onsets and compared to healthy controls [214]. Gamma-aminobutyric acid (GABA) was reduced, while levels of sphingosine 1-phosphate (S1P) were elevated [214]. Moreover, significant metabolite changes in MG were seen in pipecolic acid, 5,8-tetradecadienoic acid, bisnorcholic acid, chenodeoxycholylglycerine, coprocholic acid, coenzyme Q4, and cholylglcine [214].

Serum levels of circulating follicular helper T cells appear to be associated with disease severity and decreased after immunotherapy [215]. Similarly, calprotectin titers are increased in MG patients and are discussed to correlate with clinical manifestations [216]. Nevertheless, it must be noted that these parameters have found limited incorporation into routine assessment practices.

In anti-AChR- and anti-MuSK-Ab-positive MG cases, several elevated circulating miRNAs were found (Table 1). Interestingly, high miR-30e-5p levels are associated with risk of generalization in ocular MG [129]. Further studies of circulating miRNAs in agrin and anti-LRP4-Ab-positive MG are still required. Importantly, detected miRNAs in anti-AChR- and anti-MuSK-Ab-positive MG appear to correlate with treatment response [129,217]. Hence, further investigation may help in predictive estimations of affected patients.

Seronegative MG cases might be detected as such due to insufficiency of current standard tests [218,219]. Final categorization as seronegative MG should be held back to non-immunosuppressed patients. Efforts to establish criteria for seronegative MG subgroups aim to improve early recognition and tailored therapeutic approaches. Additionally, the expanding knowledge of different antibodies, including junctional and non-junctional types, offers potential markers for treatment response prediction [220].

The lack of specificity and the limited availability of well-established prognostic biomarkers pose significant challenges in MG. Despite previous discussions suggesting that serum autoantibody titers, particularly anti-AChR Abs, could serve as indicators of disease severity, recent studies have cast doubt on their reliability [221,222].

A notable study by Obaid et al. employed a flow cytometric approach to explore the correlation between anti-AChR-Ab titers, MAC formation, and the Myasthenia gravis composite score (MGC). Surprisingly, their findings revealed inconsistencies in a subset of patients, as there was no significant correlation observed between MAC formation, anti-AChR-Ab levels, and disease severity [91].

These discrepancies may stem from several factors, including the heterogeneity of anti-AChR Abs and the absence of standardized measurement techniques. Furthermore, variations in antibody pathogenicity add complexity to the use of anti-AChR antibody levels as reliable biomarkers for individual patients. These limitations underscore the need for further research to identify more specific and robust prognostic biomarkers for MG.

## 5. Biomarkers in IIM

### 5.1. Current Biomarkers in IIM

Idiopathic inflammatory myopathies (IIM) are a rare heterogenous cluster of autoimmune-mediated diseases affecting mainly skeletal muscles. Alongside typical manifestations with muscle weakness and fatiguing, IIMs are often accompanied by specific organ manifestations, including skin and lungs, among others. IIMs can be subclassified into different groups—dermatomyositis (DM), polymyositis (PM), immune-mediated necrotizing myopathy (IMNM), antisynthetase syndrome (ASyS), inclusion body myositis (IBM), and overlap myositis (OM) [223]. Importantly, clinical presentations, treatment responses, and prognoses differ strongly throughout subgroups [224]. Several supporting biomarkers, almost all of them auto-Abs, have been identified in the past and serve understanding causative mechanisms (Table 1). Nevertheless, IIMs are still deeply underdiagnosed.

Non-specific muscle enzymes including creatine kinase (CK), aspartate aminotransferase (AST), alanine aminotransferase (ALT), lactate dehydrogenase (LDH), and aldolase are elevated due to muscle damage and do not necessarily correlate with clinical severity or disease activity [225]. Due to the challenges involved in accurately assigning biomarkers to specific subgroups in IIMs, we have chosen to focus separately on myositis-specific Abs (MSAs) and further biomarkers of interest. The respective classifications can be found in Table 1, delineating the different categories.

### 5.2. Myositis-Specific Autoantibodies (MSA)

Some of the following MSAs can be supportive in the diagnostic workup, but their prevalence ranges only around 20–50% [134,226,227]. Detection of more than one MSA in a single patient is rather rare, though has been described before [228]. In addition to MSAs, the literature also describes myositis-associated Abs (MAAs). However, in contrast, MAAs exhibit less disease specificity, as they are found in other systemic autoimmune rheumatic diseases as well and are often associated with conditions of disease overlap [134,229].

Typical MSAs identified in patients with DM are anti-Mi-2 Abs, anti-aminoacyl-tRNA synthetase (ARS) Abs, anti-nuclear matrix protein 2 (anti-NXP2) Abs, Abs against transcription intermediary factor 1γ (TIF1γ/α), and anti-small ubiquitin-like modifier activating enzyme Abs (Anti-SAE) [133,154,155]. Abs against Mi-2, NXP2, and SAE appear to correlate with disease activity in DM. Moreover, their sequential testing may be supportive for predictive purposes [133]. Anti-Jo-1 Abs are the most frequent Abs reported among the group of anti-ARS Abs (around 15–30%). Their presence is associated with a better response to rituximab [230]. Notably, patients with non-Jo-1-anti-tRNA-synthetase Abs display reduced survival rates compared to anti-Jo-1-positive cases [137]. Anti-Mi-2 Abs also indicate a better treatment response and a favorable outcome, although they have also been sporadically detected in PM patients [131]. In addition, it is known that type I interferons are increased in patients with DM and Janus kinase inhibition improved clinical status in a proof-of-concept study [148]. Besides that, Anti-NXP2 Abs are conjoined to a specific clinical phenotype (in particular calcinosis cutis) and cancer development, showing diagnostic and prognostic potential [143,231]. Abs to TIF1γ/α are equally associated with coincident malignancy and poorer outcome [149]. As a biomarker, TIF1γ/α could aid identifying cancer-associated IIM and influence disease management. In terms of clinically amyopathic DM (CADM), anti-MDA5 Abs have a diagnostic value and are associated with vasculopathic skin ulcerations, poor prognosis, and high prevalence of interstitial lung disease (ILD) [232]. Ab titer levels appear to be positively correlated with disease severity and outcome [146]. Serial monitoring could contribute to sooner recognition of remission or relapse [147].

Patients with ASyS, characterized by a clinical syndrome of myositis, arthritis, ILD, mechanics hands, and Raynaud’s phenomenon often express anti-ARS Abs [233]. Due to ILD being more present in ASyS than in other IIMs, detection should arouse alertness for complicative clinical developments [138].

In IMNM, anti-signal recognition particle (SRP), Abs are utilized as serological indicators [156]. Correspondingly, seral 3-hydroxy-3-methylglutaryl-coenzyme A reductase (HMGCR) auto-Abs were detected in patients with IIM and particularly with IMNM [159,234]. Statins are known to potentially cause induction of anti-HMGCR in a subset of HMGCR-IMNM [158].

In seropositive IBM cases, anti-cytosolic 5′-nucleotidase 1A (cN1A) Abs were identified and appear to have a high specificity and rather moderate sensitivity but lack correlation with clinical severity [169].

### 5.3. Further Biomarkers

Biomarkers for macrophage activation can also serve as meaningful indicators with pathogenetic relevance in disease management. Serum soluble CD163 is discussed to be a predictive biomarker in PM and DM, especially in anti-MDA5 Ab positive cases. Here, titers decreased significantly after treatment [167]. Similarly, serum soluble CD206 levels exhibited a notable increase in patients with DM and showed a correlation with the presence of ILD [235]. Heightened serum neopterin levels were notably linked to rapidly progressive ILD and decreased survival among individuals with DM [236].

### 5.4. Required Biomarkers in IIM

To date, prevailing biomarkers in IIMs are primarily applicable for prognostic and predictive perspectives. They also assist in guiding classification into correct IIM subgroups. For diagnostic purposes, Abs are not sufficient alone but rather support disease identification alongside other testing tools. The actual prevalence of the referred MSAs in IIMs should also be in the focus of supplementary future studies. This is especially important to define more homogeneous clinical patient cohorts. Additionally, miRNA profiling has not been conclusively investigated in IIMs. An outline of MSAs and IIIM-associated biomarkers is found in Figure 3.

Evidently, there is a tremendous deficiency regarding monitoring biomarkers. Standardized appliances to observe disease activity are required. Treatment responses and outcomes in IIMs are still disappointing [237]. Even if subcategorizing IIM types helps in terms of estimating known complications e.g., ILD, the single variants display large variances in their respective course. Further identification of underlying pathophysiological pathways could contribute to the development of target-guided treatment options.

## 6. Outlook

Inflammatory neuromuscular disorders encompass a heterogeneous group of immune-mediated diseases, each characterized by distinct underlying mechanisms that contribute to their pathogenesis. Given the current state, pathomechanisms have not been comprehensively understood. It is therefore reasonable to anticipate that additional mechanistic backgrounds and potentially promising target structures will be identified in the years to come. This also extends to the expectation that novel suitable biomarkers will be incorporated into the standard protocols for managing these conditions.

The diagnosis of GBS and CIDP poses significant challenges due to limitations in identifying and clinically integrating specific biomarkers. Consequently, a delay in the initiation of effective immunomodulatory treatments can occur. The potential consequences can be severe for affected individuals, especially in the case of GBS, as it can lead to the need for mechanical ventilation and intensive medical care to ensure adequate treatment. The employed serum cytokines, chemokines, and complement proteins have the capacity to reflect trends in disease progression; however, they lack the required specificity.

In this field, it is crucial to concentrate efforts on the identification and implementation of specific biomarkers that can be readily and easily measured in routine clinical practice. It is worthwhile to critically note that the assessment of certain promising biomarkers is restricted to specialized centers equipped with the necessary technical infrastructure and expertise. As a result, access to these biomarker assessments is not uniformly available across all clinical sites. Consequently, there may be a potential temporal delay in the implementation of disease management strategies.

When it comes to MG, there are only a handful of biomarkers that are utilized in clinical practice. Remarkably, there are hardly any predictive biomarkers established in MG. If described, they are often linked to occurrence of postoperative myasthenic crises and not MG itself [238,239]. Alongside the “classical” biomarkers, including anti-AChR, anti-MuSK, and anti-LRP4-Abs, the remaining biomarkers listed are not disease-specific and should be regarded more as “complementary biomarkers” that contribute to a deeper understanding of the pathophysiology. Metalloproteinases such as MMP-2 and MMP-9 seem to have the potential to serve as biomarkers for tracking disease severity. Both generalized MG and CIDP patients have exhibited elevated plasma levels of MMP-9 and decreased levels of MMP-2. Furthermore, there appears to be a correlation between the concentration of MMP-2 and disease severity in MG patients [240]. While these findings are preliminary, they provide a promising foundation for further research into the role of MMPs in these conditions. Overall, it becomes abundantly clear that at this stage, there are not enough biomarkers to adequately address the disease and individual courses in affected individuals, thereby complicating the clinical management of MG.

The most prominent challenge faced by IIMs does not lie in the specificity of commonly used Abs but rather in the relatively low prevalence and subsequent lack of prompt diagnosis of this group of disorders. Additionally, there is a need for the implementation of standardized, cost-effective, and rapid detection methods.

MiRNAs are currently explored extensively in multiple studies, including research on the field of inflammatory NMDs. Currently, in IIMs and CIDP, there are barely a few studies addressing miRNAs and their potential practicality. As methods are evolving, we expect miRNAs to become increasingly important as tools for personalized medicine and as sources for treatment management. However, detection of miRNAs is still far from being applied as point of care testing assets. The requirement of reliable and fast laboratory assays is still an unmet need.

## 7. Conclusions

In summary, inflammatory neuromuscular disorders represent a diverse group of immune-mediated conditions, each characterized by distinct underlying mechanisms contributing to their pathogenesis. While our current understanding of these disorders has come a long way, there is still much to uncover. The complex nature of these diseases makes the search for reliable and specific biomarkers a crucial aspect of research and clinical practice.

Diagnosing the discussed conditions presents unique challenges. The limitations in identifying and effectively integrating specific biomarkers into clinical practice can lead to delays in initiating appropriate treatments. These delays can have severe consequences for affected individuals, particularly in cases such as GBS, where mechanical ventilation and intensive medical care may be required.

Currently, serum cytokines, chemokines, and complement proteins are used to track disease progression, but they lack the requisite specificity needed for precise diagnosis and tailored treatment. Therefore, it is imperative to focus research efforts on identifying and implementing readily measurable, specific biomarkers that can be easily incorporated into routine clinical practice. This would ensure broad accessibility and timely implementation of effective disease management strategies.

In the realm of MG, the scarcity of biomarkers hampers our ability to comprehensively understand and manage the disease. While a few biomarkers are used in clinical practice, predictive biomarkers for MG itself are notably absent. Promising biomarkers, such as microRNAs (miRNAs), are emerging, but their practical application for diagnosis and management is still in its infancy.

For IIMs, the primary challenge lies not in the specificity of commonly used antibodies but rather in their relatively low prevalence and the subsequent delay in diagnosis. Moreover, there is a pressing need for the implementation of standardized, cost-effective, and rapid detection methods to improve the diagnosis and management of these disorders.

The potential of miRNAs, although still in the early stages of practical application, is an exciting development in the field of inflammatory neuromuscular disorders. As detection methods for miRNAs evolve, they are expected to become valuable tools for personalized medicine and treatment management. However, the development of reliable and rapid laboratory assays for miRNA detection remains an unmet need.

In conclusion, ongoing research and advancements in biomarker discovery and detection methods hold the potential to significantly improve the diagnosis and management of these complex autoimmune neuromuscular disorders. As we continue to unravel the intricacies of these diseases and identify specific, easily measurable biomarkers, we move closer to enhancing the care and outcomes for individuals affected by them. The journey ahead is challenging but promising, with the ultimate goal of improving the quality of life for those living with these conditions.

## Figures and Tables

**Figure 1 cells-12-02456-f001:**
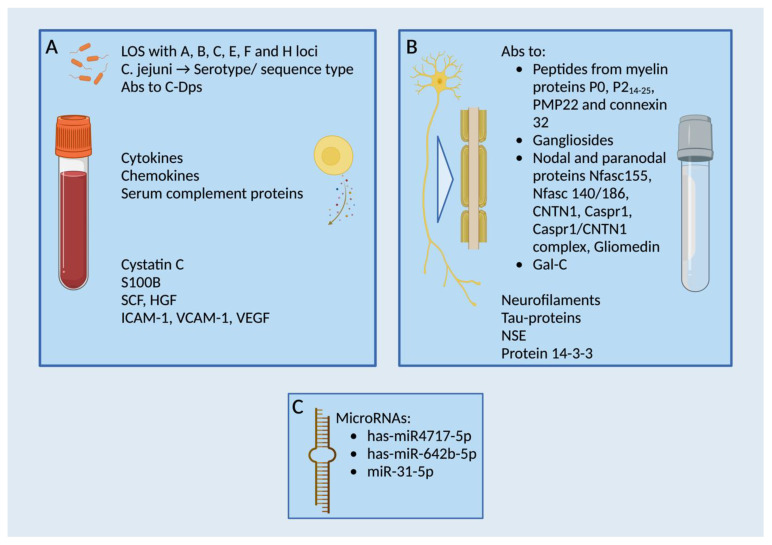
Overview of relevant biomarkers in GBS and CIDP, emphasizing pathogenetically significant locations. (**A**) Presentation mainly (but not limited to) serum biomarkers, with a focus on infection-triggered markers. (**B**) Representation primarily (but not exclusively) of biomarkers found in CSF. Of particular pathogenetic significance are structures of the myelin sheath. (**C**) Known microRNAs as potential biomarkers in GBS/CIDP. Created with BioRender.com.

**Figure 2 cells-12-02456-f002:**
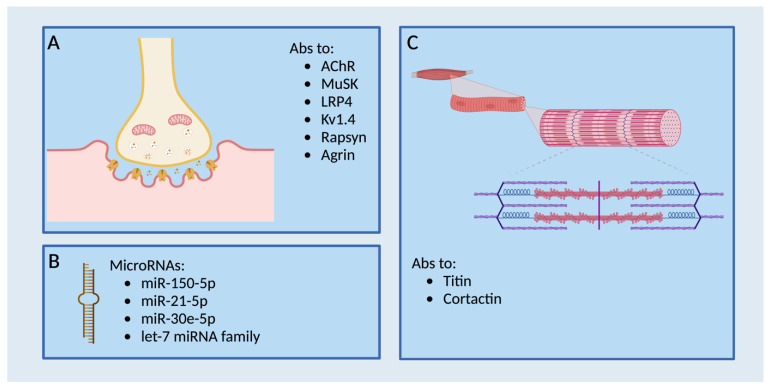
Outline of pertinent biomarkers in MG. (**A**) Significant biomarkers include antibodies associated with the NMJ. (**B**) Overview of microRNAs. (**C**) Biomarkers are not limited to the NMJ but can also play a role at the direct level of the muscle. Created with BioRender.com.

**Figure 3 cells-12-02456-f003:**
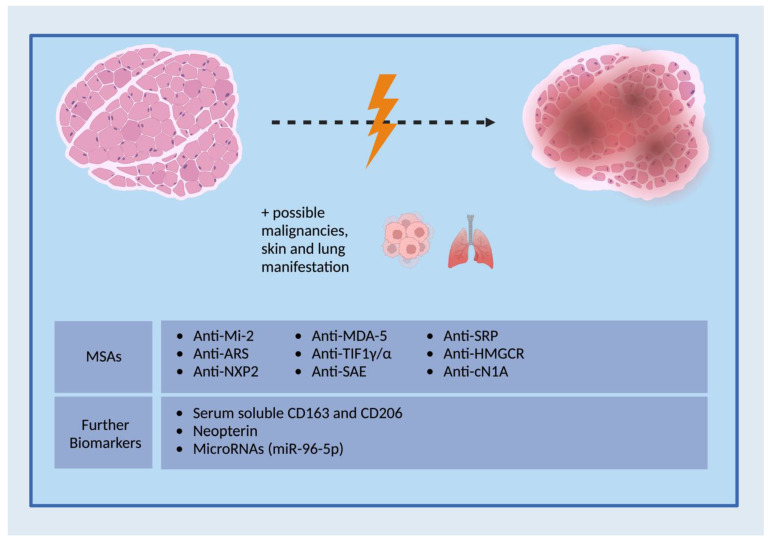
Presentation of MSAs and other biomarkers. Depending on the subclass, a specific clinical phenotype may prevail in IIMs. Created with BioRender.com.

**Table 1 cells-12-02456-t001:** Current biomarkers in autoimmune neuromuscular diseases.

Disease	Type of Biomarker	Biomarker	Detection Method	Correlation	Occurence	References	Limitations/Comment
*GBS/CIDP*	Diagnostic	Lipooligosaccharides (LOS) with A, B, C, E, F and H loci; Serotype and sequence type of *Campylobacter jejuni*	PCR screening, genes from published LOS loci and sequencingSerum and CSF	Identification of *C. jejuni*-associated GBS	GBS and Miller-Fisher syndrome (MFS)	[4,5]	
	Diagnostic	Antibodies against *Campylobacter jejuni* DNA-binding protein (C-Dps)	ELISA, Western BlotSerum	Detection of anti-C-Dps-IgG indicates *C. jejuni* related GBS	*Campylobacter jejuni*-related GBS	[6]	Also detected in patients with *C. jejuni* enteritis (rarely)
	Diagnostic	Metallo-proteinases (MMPS) (MMP-9, TIMP-1)	ELISASerum	No correlation	Described in CIDP	[7]	Not disease-specific
	Diagnostic	Antibodies to peptides from myelin proteins P0, P2_14–25_, PMP22 and connexin 32	Antigen-specific proliferation assay, Immunoprecipitation, Western BlotSerum	No correlation	Described in GBS and CIDP	[8,9,10,11,12,13,14]	
	Diagnostic Monitoring	Sphingomyelin (SM)	Fluorescence-based assayCSF	Correlation with disease activity, elevated in active CIDP.	Increased in GBS and CIDP	[15]	
	Diagnostic	Cystatin C (cysteine protease inhibitor)	ELISACSF	Decrease may be linked to higher cathepsin B activity (cathepsin B levels increased in CSF)	Significant decrease of cystatin c levels in GBS and CIDP patients	[16,17,18,19]	Decrease also observed in MS patients
	Diagnostic	Protein 14-3-3	Immunoblot assayCSF	Early detection (12 to 48 h after disease onset) in GBS	Elevated in CSF of GBS and CIDP patients	[20,21]	Not disease-specific
	Diagnostic	IL-8	Multiplex bead immunoassaysCSF	Aid in differentiation between CIDP and GBS, including acute-onset CIDP.CSF IL-8 in GBS > CIDPOptimal IL-8 cutoff → 70 pg/mL	Should be measured initially during diagnostical processHigh specificity and positive predictive value	[22]	Not disease-specific
	PredictivePrognostic	Autoantibodies to gangliosides (GM1, GA1, GD1a, GD1b, GalNAc-GD1a, 9-O-Acetyl GD1b, GD3, GM1, GT1a, GT1b, GT3, GQ1b, 0-Acetyl GT3, LM-1, GD1a/GD1b, GM1/GalNac-GD1a, GM1/PA, GM1/GD1a, GM1/GT1b, LM1/GA1)IgG and IgM	ELISASerum	Correlation with clinical phenotypes and specific symptoms of GBS e.g., ophthalmoplegia and Anti-GQ1b IgGAnti-GM1 linked to *Campylobacter jejuni*, titers correlate with clinical recovery and therapy response GM1/GalNac-GD1a linked to respiratory infection	High prevalence in GBS of Anti-GM1 and Anti-GT1a	[23,24,25,26,27,28,29]	
	PredictivePrognostic	Antibodies against nodal and paranodal proteins:Neurofascin (Nfasc155 and Nfasc140/186)Contactin-1 (CNTN1)Contactin- associated protein-1 (Caspr1) Caspr1/CNTN1 complexGliomedin	ELISA, Immunoprecipitation, cell-based AssaysSerum	Associated with specific clinical manifestation e.g., ataxia and tremorPoorer response to IVIgAnti-CNTN1 seem to benefit from corticosteroids	Nfasc155 in 4–18% of CIDP cases	[30,31,32,33,34,35,36,37,38]	
	PredictivePrognostic	Neurofilaments Phosphorylated neurofilament heavy protein (pNFH)Neurofilament light chain (Nfl)	ELISA,Electrochemiluminescence (ECL) based immunoassay Serum and CSF	Indicator for neurodegenerationPositive correlation with clinical and electrophysiological presentation in GBSAssociation with disease progression and therapy outcome in CIDP	Elevated in both serum and CSF of GBS and CIDP patients	[25,39,40,41,42,43]	General indicators of axonal damage, also detectable in other patient groups with evidence of structural CNS damage
	Predictive Prognostic	Tau-proteins	ELISACSF	Correlation with clinical manifestation and poorer clinical outcome	Elevated in CSF of GBS and CIDP patients	[25,42,44,45]	Also detectable in other patient groups with evidence of structural CNS damage e.g., Alzheimer’s Disease
	Prognostic	Autoantibodies against galactocerebroside (Gal-C)	ELISA Serum	Association with sensory deficits and autonomic disruption in GBSAssociation with Mycoplasma pneumoniae infection	GBS	[46,47]	
	Prognostic	Neuron-specific enolase (NSE)	Enzyme immunoassay methodsCSF	Higher levels correlate with a longer duration of disease	Elevated in CSF of GBS and CIDP patients	[48,49,50]	CSF-NSE is not GBS or CIDP specific; Elevation is also observed other conditions e.g., Creutzfeldt-Jakob disease [51]
	PrognosticPredictiveMonitoring	CytokinesInterferon gamma (IFN γ), Tumor necrosis factor α (TNF α), Transforming growth factor β1 (TGF β1), IL-1β, IL-4, IL-6, IL-10, IL-12, IL-16, IL-17, IL-18, IL-22, IL-23, IL-37	ELISA based assays (multiplexed fluorescent bead-based immunoassaySerum and CSF	TNF α and IFN γ are elevated in GBS and correlate with clinical severityTGF β1 levels are decreased in the early course of GBS, downregulation correlates with clinical disabilitySerum levels are positively correlated with GBS disease severity and decreased after IVIg treatment (IL-17A, IL-37)CSF IL-17A levels were positively correlated with clinical manifestation of GBSUpregulation of IL-4 and IL-10 are linked to recovery phase in GBSCSF and serum levels of interleukins declined after IVIg treatment	Cytokine elevation is described in CIDP and GBS	[52,53,54,55,56,57,58,59,60,61,62,63,64,65]	Not disease-specific.
	PrognosticMonitoring	Serum complement proteinsC3, C3a, C5a, C5b-9	NephelometrySerum	Upregulation predicts poor prognosisHigh C3 correlates with complement activation with high C3a und C5aCorrelation with disease activity	Increased in GBS and CIDP	[66,67,68]	Not disease-specific
	PrognosticMonitoring	S100B protein (calcium-binding astroglial protein)	ELISASerum and CSF	Elevated in CSF and serumAssociation with clinical severity and poor prognosisDecrease in stable disease course	Elevated in GBS and CIDP	[42,49,50]	Expression of S100B is not restricted to neural tissue; Serum levels can be increased after e.g., bone fractures or hepatic injury [69,70]
	Prognostic Diagnostic	Stem cell factor (SCF)Hepatocyte growth factor (HGF)	Multiplex bead-based ELISACSF	Increased	Elevated levels in CSFCIDP > GBSCorrelation with chronicity	[71]	Value of this examination is still uncertain
	Monitoring	ChemokinesCCR2, CCL7,CCL3, CCL27, CCR1, CCR5, CXCL10, CXCR3, CXCL9, CXCL12, monocyte chemoattractant protein 1 (MCP-1)	Multiplex bead-based ELISASerum and CSF	CCR2 and MCP-1 decreased in the recovery stage	Increased in GBS and CIDP	[25,72,73,74]	Not disease-specific
	Monitoring	Intercellular adhesion molecule 1 (ICAM-1)Vascular cell adhesion molecule 1 (VCAM-1)Vascular Endothelial Growth Factor (VEGF)	Multiplex bead-based ELISACSF	Decrease after therapy in studiesCorrelation with repair processes is currently being studied	Increased in GBS and CIDP	[71,75,76]	Not disease-specific
	Monitoring	MicroRNAshas-miR4717-5p (GBS)has-miR-642b-5p (GBS)miR-31-5p (CIDP)	Microarray, droplet digital PCRSerum	High levels of miR-31-5p correlate with longer disease durationPotential in improving personalized patient care	Detectable in GBS and CIDP	[77,78,79,80]	
*MG*	DiagnosticMonitoring	Anti-AChRs (muscle nicotinic acetylcholine receptors)IgG subtype 1 and 3	Radioimmunprecipitation assay (RIPA) with high specificity and sensitivity, fixed cell-based assaysSerum	Monitoring in patients with immunosuppressive treatmentHigher levels in ocular MG are associated with conversion to generalized MGHigher levels in late-onset MG	85% in generalized MG, highly specific for MG	[81,82,83,84,85,86,87,88,89,90,91,92,93,94,95,96,97,98,99]	Lower titers in ocular MGInconsistent studies regarding correlation with disease severity and treatment response
	DiagnosticPrognosticPredictive	Anti-MuSK (Muscle-specific kinase)IgG subtype 4	Radioimmunoprecipitation assay (RIPA), ELISA,cell-based assay (CBA)Serum	Correlation with disease severityAffection of facial-bulbar musclesEarly crises and challenging treatmentWorse outcomeAssociation with early onset MG and better response to rituximab	5–8% of MG30–50% of AChR-negative MG	[100,101,102,103,104,105,106,107,108,109,110]	More often in female patientsHighest sensitivity in detection via CBA
	Diagnostic	Anti-LRP4 (low-density lipoprotein receptor-related protein 4)IgG subtype 1 and 2	Cell-based assay (CBA)Serum	Stronger clinical manifestation than in seronegative MG	2% of MG; higher in non-AChR and non-MuSK cases	[111,112,113,114,115,116,117]	Higher prevalence in female patientsNot specific for MG (e.g., found in ALS as well)
	DiagnosticPrognostic	Anti-Titin	ELISA, Cell-based assay (CBA)Serum	Thymoma-associated MGMore frequent hospitalization	20–40% in Anti-AChR-positive MG	[95,118,119,120]	Screening for thymoma presence should follow positive testing
	DiagnosticPrognostic	Anti-Kv1.4 (voltage gated potassium channel)	Cell-based assay (CBA), Radioimmunoprecipitation assay (RIPA)Serum	Association with myasthenic crises and thymomaAssociation with bulbar manifestation, myocarditis and QT-Time prolongation (Japanese population)	11–18% (Japanese MG population)	[95,121,122]	More frequent in female patientsExpensive detection
	Diagnostic	Anti-Rapsyn	ELISASerum	No correlations	15% of MG	[95,123]	Not MG-specific
	DiagnosticPrognostic	Anti-Cortactin	Western Blot, ELISASerum	Mild symptoms	10–25% of MG	[95,124,125,126]	Not MG specific
	DiagnosticPrognosticPredictive	Anti-AgrinIgG subtype 1 and 3	ELISA; CBA; serum	Correlation with limited therapeutic response and mild to severe clinical manifestation	2–5% of MG, mainly seropositive MG	[95,127,128]	More frequent in male patients
	Diagnostic Prognostic Predictive	Micro-RNAsmiR-150-5pmiR-21-5pmiR-30e-5plet-7 miRNA family	Microarray, droplet digital PCRSerum	Correlation with treatment responseHigh miR-30e-5p levels are associated with risk of generalization in ocular MG	Studied in AchR and MuSK-positive MG	[129]	
*IIM*	DiagnosticPredictivePrognostic	Anti-Mi-2	ELISA, ImmunoblotSerum	Classical DM, associated with beneficial prognosis, mild myositis, lower risk of ILD, better treatment response especially to rituximabCorrelation with disease activityAssociation with HLA-DR7	MSA2–45% prevalence in DM	[130,131,132,133,134,135]	Positive sera may also be found by ELISA in PM patients [136]Prevalence can only be estimated (varying among different countries)
	DiagnosticPredictivePrognostic	Anti-ARS(aminoacyl-tRNA synthetases)Jo-1, PL-7, PL-12, EJ, OJ, KS, ZO, YRS	RNA-Immunoprecipitation, ELISA, Line Blots,Serum	Association with ASySHigher mortality and interstitial lung disease (ILD) incidence in non-Anti-Jo-1-ARS (+)Higher treatment dosage required	MSAAnti-Jo-1 15–30% in DM/PMOthers < 5%	[134,137,138,139,140,141]	Low prevalence of non-Anti-Jo-1-ARSRNA-Immunoprecipitation not widely availableRates of false-positive cases higher in Line Blots [140]
	DiagnosticPredictivePrognostic	Anti-NXP2 (anti-nuclear matrix protein 2)	Immunoprecipitation,Western BlotSerum	Association with calcinosis and severe myositis, cancer developmentCorrelation with disease activity	MSAAdult and juvenile DM1–5%	[134,142,143,144,145]	Immunoassays have been released and are currently discussed
	DiagnosticPredictivePrognosticMonitoring	Anti-MDA-5(Melanin differentiation-associated protein-5)/CADM140	Immunoprecipitation, Western Blot, ELISASerum	Associated with clinically amyotrophic DM (CADM), ILD, poor prognosis, sever skin manifestationTiter levels linked to disease severity and outcome	MSA15–20% in IIM, mainly CADM	[133,134,146,147]	Higher prevalence in AsiaMore frequent in women
	DiagnosticPrognostic	Anti-TIF1γ/α (transcription factor 1 γ/α)	ELISA, ImmunoprecipitationSerum	Malignancy-associated DM	MSA10–15%, higher prevalence in cancer-associated DM, rare in PM	[134,148,149,150,151,152]	Cancer association is applied to adults [151]
	DiagnosticPredictivePrognostic	Anti-SAE (small ubiquitin-like modifier activating enzyme)	Immunoprecipitation,Indirect Immunofluorescence testSerum	Cancer associationSerum levels correlate with disease activity	MSA1–5% in DM	[153,154,155]	
	DiagnosticPrognostic	Anti-SRP (Anti-signal recognition particle)	RNA Immunoprecipitation, ELISASerum	Associated with a rapidly progressive disease course with severe weaknessCancer-associated SRP-IMNM	MSA20–25% in IMNM	[156,157,158]	More frequent in womenPrimarily in adults
	DiagnosticPrognostic	Anti-HMGCR (3-hydroxy-3-methylglutaryl-coenzyme A reductase)	Immunoprecipitation,ELISASerum	Significant association with serum creatine kinaseHigher serum muscle enzymes than in other IIMCorrelation with disease activityCancer association	MSA6–12% in IIM	[158,159,160,161,162,163,164,165]	Cave: Statin therapy!
	PredictivePrognostic	Serum soluble CD163	ELISASerum	Biomarker for macrophage activationCorrelation with disease severityAssociation with Anti-MDA5 (+) cases	PM/DM	[166,167,168]	Not IIM-specific; supporting
	Diagnostic	Anti-cN1A (cytosolic 5′-nucleotidase 1A)	Addressable laser bead immunoassay (ALBIA), ELISASerum	No correlation with disease severity	Around 50% in IBM	[169,170]	Moderate sensitivity, high specificity in ALBIA [171]Not IBM-specific, found also in known autoimmune diseases e.g., SLE [172]
	Diagnostic	Micro-RNAsmiR-96-5p	RTqPCRSerum	No correlation described	Upregulation in PM, DM and Anti-Jo1 positive cases	[173]

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
