# Peer review of "Current Biomarker Strategies in Autoimmune Neuromuscular Diseases"

_cells, 2023, doi:10.3390/cells12202456_

Round 1
Reviewer 1 Report
This is an excellent review article where Oeztuerk et al provided up-to-date information regarding bio-markers in autoimmune neuromuscular disorders, particularly GBS/CIDP, myasthenia gravis, and inflammatory myositis.
A Summary table providing different subtypes of biomarkers including diagnostic, prognostic, and monitoring was very helpful. Moreover, after each section, an illustrative figure summarizing the pertinent subtypes of biomarkers was also very illuminating.
This review article provides a excellent framework for future studies.
Critique: While the utility of most of the biomarkers is evident; for others, a rationale was provided by the authors, however for some, the association was not clear. This includes cysteine protease inhibitor cystatin C. Perhaps the authors can provide some additional details on this biomarker. Otherwise, the article was nicely written and very high yield.
Author Response
Response to Reviewer 1 Comments
- Summary
Thank you very much for your thoughtful and constructive feedback on our review article. We greatly appreciate your positive comments regarding the up-to-date information and the usefulness of the summary table and illustrative figures. We also acknowledge your critique and will certainly take your suggestion to provide additional details on the biomarker cysteine protease inhibitor cystatin C into consideration. Your feedback is invaluable to us in improving the quality and comprehensiveness of our work.
- Point-by-point response to Comments and Suggestions for Authors
Comment 1: Critique: While the utility of most of the biomarkers is evident; for others, a rationale was provided by the authors, however for some, the association was not clear. This includes cysteine protease inhibitor cystatin C. Perhaps the authors can provide some additional details on this biomarker. Otherwise, the article was nicely written and very high yield.
Response 1: We thank the reviewer for bringing up this very valid point. Therefore, we included additional details and updated our table (page 13, lines 133-137)
Reviewer 2 Report
This review article explores the role of several biomarkers in inflammatory neuromuscular disorders. The topic is very important, as new therapies are coming out one after another. Overall, the article is well-written and the figures are useful. I find it very interesting, but I have some concerns:
CIDP
· The pathophysiology of CIDP is very complex and involves both T and B lymphocytes as well as innate immunity. The authors described the role of IFN-gamma and TNF-alfa, but there was no reference to metalloproteinases. It has been recently reported a role for MMPs in the pathogenesis of CIDP patients, discuss and suggest their appropriate collocation among discussed biomarkers (Expression pattern of matrix metalloproteinases-2 and -9 and their tissue inhibitors in patients with chronic inflammatory demyelinating polyneuropathy. Neurol Sci. 2021 May 14. doi: 10.1007/s10072-021-05314-y).
Myasthenia Gravis (MG):
· Biomarkers in MG are essential to monitor the disease course as well as predict crises, fluctuations, and clinical deteriorations. The author clearly classified MG patients according to the antibody type. This is very important to approach with the correct treatment: this is the case of antiCD20 therapies which are more useful in antiMuSK MG compared with AChR. I suggest reading and citing a review article showing the effect of rituximab in AChR MG (Rituximab in AChR subtype of myasthenia gravis: systematic review. JNNP 2020).
· Lines 390-393. MuSKAbs correlate with disease severity; however, they are IgG4, hence they do not activate complement cascade. Hence their presence might be considered a biomarker of response to antiCD20, but not to complement inhibitors.
· Lines 400-401. AChR Abs are among IgG1 and 3 subgroups: this paves the use of FcRn-antagonists for disease treatment. From this perspective, serum IgG should be useful as a biomarker for response to treatment as well as AChRAb titers. A recent study demonstrated a relationship between AChRAb titers, serum total IgG, and severity of the disease in MG (Efgartigimod beyond myasthenia gravis: the role of FcRn-targeting therapies in stiff-person syndrome. J Neurol. 2023 Sep 8. doi: 10.1007/s00415-023-11970-1).
· Lines 436-437. Also, AChR, MuSK, or double-seronegative MG differ in pathophysiology and treatment: read and cite a recent review (Diagnosis and Management of Seronegative Myasthenia Gravis: Lights and Shadows. Brain Sci 2023).
· Recently, the role of serum light chain has been proposed as a biomarker in MG (The role of serum free light chain as biomarker of Myasthenia Gravis. Clin Chim Acta. 2022 Mar 1;528:29-33. doi: 10.1016/j.cca.2022.01.004). I think it is worth mentioning.
· Finally, MMP-3 and MMP-9 have been studied also in generalized MG and seem to be related to the severity of the disease. Discuss and cite recent literature (Metalloproteinases and Tissue Inhibitors in Generalized Myasthenia Gravis. A Preliminary Study. Brain Sci. 2022 Oct 26;12(11):1439. doi: 10.3390/brainsci12111439). These are still preliminary results but should be included in “further biomarkers” or “future directions”.
Minor points:
· title: I suggest removing “strategies”.
· References 4 and 6 are quite old (2014-2015). I suggest citing a more recent review of the literature on inflammatory polyneuropathies (Inflammatory polyradiculoneuropathies: Clinical and immunological aspects, current therapies, and future perspectives. European Journal of Inflammation. 2020;18. doi:10.1177/2058739220942340).
minor mistakes
Author Response
Response to Reviewer 2 Comments
- Summary
We would like to thank the reviewer for the time and attention invested in the topic. The insights and feedback they shared have proven to be exceptionally beneficial to us.
- Point-by-point response to Comments and Suggestions for Authors
Comment 1: The pathophysiology of CIDP is very complex and involves both T and B lymphocytes as well as innate immunity. The authors described the role of IFN-gamma and TNF-alfa, but there was no reference to metalloproteinases. It has been recently reported a role for MMPs in the pathogenesis of CIDP patients, discuss and suggest their appropriate collocation among discussed biomarkers (Expression pattern of matrix metalloproteinases-2 and -9 and their tissue inhibitors in patients with chronic inflammatory demyelinating polyneuropathy. Neurol Sci. 2021 May 14. doi: 10.1007/s10072-021-05314-y).
Response 1: We thank the reviewer for drawing our attention to MMPs and therefore have added an additional paragraph to this point (page 13, lines 113-120). We have also included the proposed review as a citation.Furthermore, we have incorporated the suggested biomarker into our table (page 3).
Comment 2: Biomarkers in MG are essential to monitor the disease course as well as predict crises, fluctuations, and clinical deteriorations. The author clearly classified MG patients according to the antibody type. This is very important to approach with the correct treatment: this is the case of antiCD20 therapies which are more useful in antiMuSK MG compared with AChR. I suggest reading and citing a review article showing the effect of rituximab in AChR MG (Rituximab in AChR subtype of myasthenia gravis: systematic review. JNNP 2020).
Response 2: We thank the reviewer for recommending this interesting article and have accordingly included the results presented in our review and in the provided table (page 19, 4.3, lines 400-402).
Comment 3: Lines 390-393. MuSKAbs correlate with disease severity; however, they are IgG4, hence they do not activate complement cascade. Hence their presence might be considered a biomarker of response to antiCD20, but not to complement inhibitors.
Response 3: We thank the reviewer for bringing up this very valid point. Therefore, we have included a paragraph addressing this concern at the appropriate place (page 19, 4.4., lines 409-411)
Comment 4: Lines 400-401. AChR Abs are among IgG1 and 3 subgroups: this paves the use of FcRn-antagonists for disease treatment. From this perspective, serum IgG should be useful as a biomarker for response to treatment as well as AChRAb titers. A recent study demonstrated a relationship between AChRAb titers, serum total IgG, and severity of the disease in MG (Efgartigimod beyond myasthenia gravis: the role of FcRn-targeting therapies in stiff-person syndrome. J Neurol. 2023 Sep 8. doi: 10.1007/s00415-023-11970-1).
Response 4: We agree with the reviewer and cited the mentioned article regarding the use of IgG and AChR Abs as biomarkers for response to treatment in MG patients (page 20, 4.5, lines 424-427). In our opinion, the point raised is highly valuable in helping readers gain a better understanding of the pathogenesis.
Comment 5: Lines 436-437. Also, AChR, MuSK, or double-seronegative MG differ in pathophysiology and treatment: read and cite a recent review (Diagnosis and Management of Seronegative Myasthenia Gravis: Lights and Shadows. Brain Sci 2023).
Response 5: We have read and cited the above mentioned review on the management of seronegative MG (page 20, 4.6, lines 462-465).
Comment 6: Recently, the role of serum light chain has been proposed as a biomarker in MG (The role of serum free light chain as biomarker of Myasthenia Gravis. Clin Chim Acta. 2022 Mar 1;528:29-33. doi: 10.1016/j.cca.2022.01.004). I think it is worth mentioning.
Response 6: We thank the reviewer for highlighting this aspect and have added a paragraph on the potential of serum light chain as a biomarker in MG (page 19, 4.2, 391-395). Since free serum light chains (FLC) are generally considered nonspecific and mainly serve a diagnostic-supporting role, we mentioned them in our discussion but opted not to include them in the table.
Comment 7: Finally, MMP-3 and MMP-9 have been studied also in generalized MG and seem to be related to the severity of the disease. Discuss and cite recent literature (Metalloproteinases and Tissue Inhibitors in Generalized Myasthenia Gravis. A Preliminary Study. Brain Sci. 2022 Oct 26;12(11):1439. doi: 10.3390/brainsci12111439). These are still preliminary results but should be included in “further biomarkers” or “future directions”.
Response 7: We mentioned these preliminary results in our Outlook section (page 24, 6, lines 603-609). We also cited the study mentioned above. Thank you very much for providing these very beneficial insights.
- Minor points:
Point 1 title: I suggest removing “strategies”
Response 1: We appreciate the feedback from the reviewer and would like to emphasize that the title 'Current Biomarker Strategies in Autoimmune Neuromuscular Diseases' was carefully chosen to reflect the holistic approach of our work. Our research encompasses not only classical antibodies but also a wide range of biomarker components, including the complement system and chemokines, comprehensively. This title was selected to highlight the innovative and comprehensive nature of our study as it encompasses various strategies for identifying and applying biomarkers in the research of autoimmune neuromuscular diseases. We believe this title accurately captures the diversity and depth of our work and hope that it can be retained.
Point 2: References 4 and 6 are quite old (2014-2015). I suggest citing a more recent review of the literature on inflammatory polyneuropathies (Inflammatory polyradiculoneuropathies: Clinical and immunological aspects, current therapies, and future perspectives. European Journal of Inflammation. 2020;18. doi:10.1177/2058739220942340).
Response 2: We thank the reviewer for recommending this more appropriate review and have added it at to the suitable place (line 81, line 84).
Round 2
Reviewer 2 Report
The authors have addressed all my concerns. The manuscript is highly improved. No further comments.